# Minocycline Inhibits Tick-Borne Encephalitis Virus and Protects Infected Cells via Multiple Pathways

**DOI:** 10.3390/v16071055

**Published:** 2024-06-29

**Authors:** Mengtao Cao, Wei Yang, Jintao Yang, Yanli Zhao, Xiaoyu Hu, Xiaoli Xu, Jing Tian, Yue Chen, Hongxia Jiang, Ruiwen Ren, Chunyuan Li

**Affiliations:** 1Center for Disease Control and Prevention of Southern Theater Command, Guangzhou 510507, China; caomengtao2021@163.com (M.C.); rakie23@163.com (X.H.); tianjing1980s@126.com (J.T.); redcellchen@163.com (Y.C.); 2Guangdong Provincial Key Laboratory of Veterinary Pharmaceutics Development and Safety Evaluation, College of Veterinary Medicine, South China Agricultural University, Guangzhou 510642, China; 15776581772@163.com (W.Y.); 20211027013@stu.scau.edu.cn (J.Y.); zyl1826876555@outlook.com (Y.Z.); hxjiang@scau.edu.cn (H.J.); 3College of Veterinary Medicine, South China Agricultural University, Guangzhou 510642, China; 4Instrument Analysis & Research Center, South China Agricultural University, Guangzhou 510642, China; xuxiaoli@scau.edu.cn; 5Guangdong Arbovirus Diseases Emergency Technology Research Center, Guangzhou 510507, China

**Keywords:** TBEV, minocycline, calcium ion, MAPK-ERK, IL-6

## Abstract

Tick-borne Encephalitis (TBE) is a zoonotic disease caused by the Tick-borne Encephalitis virus (TBEV), which affects the central nervous system of both humans and animals. Currently, there is no specific therapy for patients with TBE, with symptomatic treatment being the primary approach. In this study, the effects of minocycline (MIN), which is a kind of tetracycline antibiotic, on TBEV propagation and cellular protection in TBEV-infected cell lines were evaluated. Indirect immunofluorescence, virus titers, and RT-qPCR results showed that 48 h post-treatment with MIN, TBEV replication was significantly inhibited in a dose-dependent manner. In addition, the inhibitory effect of MIN on different TBEV multiplicities of infection (MOIs) in Vero cells was studied. Furthermore, the transcriptomic analysis and RT-qPCR results indicate that after incubation with MIN, the levels of TBEV and *CALML4* were decreased, whereas the levels of calcium channel receptors, such as *RYR2* and *SNAP25*, were significantly increased. MIN also regulated MAPK-ERK-related factors, including *FGF2*, *PDGFRA*, *PLCB2*, and p-ERK, and inhibited inflammatory responses. These data indicate that administering MIN to TBEV-infected cells can reduce the TBEV level, regulate calcium signaling pathway-associated proteins, and inhibit the MAPK-ERK signaling pathway and inflammatory responses. This research offers innovative strategies for the advancement of anti-TBEV therapy.

## 1. Introduction

Tick-borne Encephalitis (TBE) is an acute infectious disease of the central nervous system (CNS) of humans and animals caused by infection by the Tick-borne Encephalitis virus (TBEV) of the *Orthoflavivirus* genus [1]. There is an increasing incidence of emerging/re-emerging, neurotropic viruses, including TBEV, all around the world [2]. Approximately 10,000 cases of TBEV infection are reported worldwide annually, presenting a significant public health concern [3]. Research has demonstrated that these viruses disrupt intracellular calcium ion homeostasis upon invading a cell by manipulating the calcium signaling pathway. Dengue virus (DENV) and Zika virus (ZIKV), which also belong to the *Orthoflavivirus* genus, can activate calcium channels to promote viral replication [4]. In a DENV-infected mouse model, DENV induced the phosphorylation of ERK1/2 and heightened the degree of apoptosis in the hepatocytes, facilitating DENV replication [5]. However, changes in the intracellular calcium level and MAPK-ERK signaling pathway after TBEV infection have not been reported. In addition, TBEV infection induces innate immune responses in mice and human neuronal cells and increases the expression levels of cytokines such as IL-6 [6]. At present, there have been different advances in prevention and treatment options for TBEV. The prevention of TBE primarily involves two approaches: non-specific measures and specific immunization. Non-specific precautions include pasteurizing milk, reducing tick populations, and personal protection [1]. Specific prevention through active immunization is widely regarded as the most effective method against TBE. Various vaccines like Encepur N, FSME-Immun CC, Ticovac, Encevir-Neo, and Klesh-E-Vak are commonly used and have shown high efficacy in TBE prevention [7,8]. However, there have been several confirmed cases of TBE, which developed despite vaccination [9,10,11]. As there are no specific antivirals approved for the treatment of TBE, specific therapeutic agents and strategies are needed for the treatment of unvaccinated patients and vaccinated individuals with post-vaccine complications and breakthrough TBE [12]. Therefore, it is an urgent issue for us to search for specific anti-TBEV drugs.

Several studies have demonstrated that tetracycline antibiotics possess properties beyond antimicrobial activity. Specifically, their efficacy as antiviral agents has been experimentally and clinically proven [13]. Minocycline (MIN), due to its high fat solubility and long-acting nature, is able to penetrate the blood-brain barrier, resulting in higher concentrations in the brain tissue compared to other tetracycline antibiotics [14]. The primary clinical symptoms of TBE are neurological disorders affecting the brain, indicating that minocycline could potentially inhibit the replication of TBEV. While minocycline’s antiviral properties have been extensively studied and it has been proven to be effective against various viruses in the *Orthoflavivirus* genus, such as West Nile virus (WNV) [15], Japanese encephalitis virus (JEV) [16], and DENV [17], its specific therapeutic impact on TBEV infection has not yet been reported.

This study investigated the impact of MIN on TBEV replication and elucidated its initial mechanism of action. Our findings suggested that MIN effectively inhibited TBEV replication in a dose-dependent manner. The inhibition of viral replication by MIN is attributed to the modulation of calmodulin and calcium channel-associated proteins. Moreover, MIN maintained normal cellular differentiation and value addition, mitigated cellular damage induced by TBEV infection, and suppressed the MAPK-ERK signaling pathway to impede TBEV replication. Furthermore, MIN reduced the inflammatory response to TBEV infection by down-regulating the expression of the inflammatory factor IL-6.

## 2. Materials and Methods

### 2.1. Cell Culture and Virus Propagation

Vero (KCB92017YJ) and Madin-Darby canine kidney (MDCK) cells (KCB2006105YJ) obtained from Kunming Cell Bank (Academy of Sciences, Kunming, China) and Lilly Laboratories Cell-Monkey Kidney 2 (LLC-MK2) cells obtained from CCTCC (Wuhan, China) were cultured in Dulbecco’s modified Eagle medium (DMEM, Gibco, CA, USA) supplemented with 10% fetal bovine serum (FBS, Sigma-Aldrich, MO, USA) in a 37 °C humidified atmosphere with 5% CO_2_. The TBEV strain (GenBank: KU885457.1, European subtype) used in this research was maintained by the Emergency Technical Research Center for Insect-borne Viral Diseases in Guangdong Province. There were few differences between the sequences obtained by TBEV sequencing (Appendix A) and the provided Genebank sequences. The TBEV was cultured in the Vero, MK2, and MDCK cells until cytopathy was observed. The infected cells were subjected to three freeze-thaw cycles to release the viral particles into the medium. The medium was subsequently centrifuged at 10,000 g for 10 min to remove the cellular debris and filtered through a 0.22 µm filter (Merck, Darmstadt, Germany) to obtain the viral supernatant. The level of TBEV reproduction was measured using RT-qPCR; a low Ct value of the Vero cells indicates successful infection (Appendix A). The viral titers were determined using a 50% tissue culture infectious dose (TCID_50_) assay on the Vero cells, and the viruses were stored at −80 °C until required.

### 2.2. Preparation of Compounds

Minocycline (A600356, Sangon Biotech, Shanghai, China) and Amantadine salt (AMT, A800770, MACKLIN, Shanghai, China) were dissolved in sterile water to concentrations of 20 mM and 10 mM, respectively. Doxycycline hydrochloride (DOX, D832390, MACKLIN, China) and Tetracycline (TET, T829835, MACKLIN, China) were dissolved in DMSO to 20 mM. The diluted drugs were filtered through a 0.22 µm filter and stored in separate packs away from light at −80 °C.

### 2.3. Cytotoxicity of Compounds

To assess the cytotoxicity of MIN, DOX, TET, and AMT, the Vero cells were seeded into 96-well plates and incubated for 12 h. Subsequently, the cells were treated with various concentrations of the compounds for 72 h. Cell viability was determined using the Cell Counting Kit-8 (C0038, Beyotime, Shanghai, China) following the manufacturer’s protocol. Absorbance at 450 nm and a reference wavelength of 650 nm were measured using a Multimode Microplate Reader (TECAN, Männedorf, Switzerland).

### 2.4. Measurement of Virus Titers

The Vero cells were inoculated into 96-well plates and cultured for 12 h. Subsequently, the cells were infected with TBEV (10TCID_50_) for 2 h. Different dilutions of the indicated compounds were then added at a volume of 100 μL/well, with six replicate wells for each dilution. The supernatant was harvested at 24, 48, and 72 h post-treatment, which was then diluted 10-fold using DMEM, and dilutions ranging from 10^−1^ to 10^−9^ were used to infect Vero cells for 2 h. A control group of uninfected normal cells was included, and six experimental wells were designated for each dilution gradient. Following virus infection, each well was supplemented with DMEM containing 2% FBS, and then cultured for 6 days. Cell growth and the cytopathic effects (CPEs) were monitored at 24 h intervals. The TCID_50_ values were determined for various concentrations of the compounds at different time points using the Reed-Muench method [18].

### 2.5. Indirect Immunofluorescence (IFA) Assay

The Vero cells were washed three times with PBS, followed by fixing with 4% paraformaldehyde at room temperature (RT) for 20 min. Subsequently, the cells were subjected to permeation for 5 min at RT with diluted 0.3% TritonX-100 (P0096, Beyotime, China). To prevent non-specific binding, 5% bovine serum albumin (BSA) was incubated for 2 h at RT. After blocking, a primary antibody specific to TBEV was added to each well (1:300 in PBS; our laboratory provided this) and incubated overnight at 4 °C. The cells were washed three times with PBST, and then incubated with a FITC-labeled goat anti-mouse fluorescent secondary antibody (1:300 in PBS, A-11029, Invitrogen, CA, USA) for 2 h at RT. After incubation, the cells were washed three times with PBST and stained with a DAPI staining solution (C1005, Beyotime, China), followed by 5 min incubation at RT. Finally, the cells were observed using a fluorescence inverted microscope, and images (Zeiss, Oberkochen, Germany) were captured to document the results.

### 2.6. Western Blots

Cell lysis was performed using RIPA buffer (P0013C, Beyotime, China) with a protease-phosphatase inhibitor (P1045, Beyotime, China) to extract the total protein. The protein concentrations were determined using a BCA kit (T9300A, TaKaRa, Kyoto, Japan) following the manufacturer’s instructions. Subsequently, the processed samples were separated using 10% SDS-PAGE and transferred to nitrocellulose membranes using a semi-dry blotting system (PB0012, ThermoFisher Scientific, Waltham, MA, USA). The membrane was blocked using 5% skimmed milk in 1× Tris-buffered saline containing 0.1% Tween 20 (TBST) at RT for 2 h, and then incubated with a primary antibody against TBEV (1:500; our laboratory provided this), GAPDH (1:1000, 5174S, CST, Boston, MA, USA), Phospho-ERK1/2 (1:1000, 4695S, CST, USA), or ERK (1:1000, 4377S, CST, USA) at RT for 1.5 h. The membranes were washed three times with TBST for 8 min each; this was followed by incubation with HRP-labeled goat anti-mouse or goat anti-rabbit secondary antibodies at RT for 2 h. Subsequently, the protein bands were visualized using a chemiluminescence imaging system (Bio-Rad, Hercules, CA, USA) with an ultrasensitive chemiluminescent solution (SQ201 EpiZyme Bio, Shanghai, China).

### 2.7. Real-Time Reverse Transcription PCR (RT-qPCR)

The total RNA of the sample was extracted using the RNAiso Plus kit (9109, TaKaRa, Japan), and then subjected to first-strand cDNA synthesis with StarScript Ⅲ RT Master Mix (A233-10, GenStar, Beijing, China) according to the manufacturer’s protocol. RT-qPCR was performed with the TB Green Fast qPCR Mix (RR430A, TaKaRa, Japan) and a One-Step RT-qPCR Kit (Probe, ER101, SANSHIBIO, Shijiazhuang, China) using the CFX96 System (Bio-Rad, USA). The specific primers [19] and PCR amplification parameters are shown in Appendix A, Appendix A and Appendix A, respectively. The expression level of the target gene was normalized against the GAPDH or PSMB4 gene using the comparative Ct method (2^−ΔΔCt^).

### 2.8. Transcriptome Sequencing

RNA sequencing was performed on the Vero cells after the different treatments, and a total of 4 groups were set up with 3 replicates in each group to determine the changes in their mRNA expression profiles. Vero cells were infected with 10TCID_50_ TBEV for 2 h, treated with 20 µM MIN for 48 h, and then total RNA was extracted. The obtained RNA samples were sent to Sangon Biotech for RNA sequencing. These datasets were deposited at the National Institute of Health’s sequence read archive under the BioProject PRJNA1095166. The R language program was used to further analyze the sequencing results to find the differentially expressed genes (DEGs). The protein-protein interaction (PPI) analysis of the DEGs was performed using the STRING database (https://cn.string-db.org, accessed on 21 November 2023). Subsequently, the DAVID Bioinformatics online analysis database (https://david.ncifcrf.gov, accessed on 21 November 2023) was used to perform gene ontology (GO) biological function enrichment and Kyoto gene and genome encyclopedia (KEGG) signal pathway enrichment analysis. Data processing and visualization were performed using the microorganism online analysis platform (https://www.bioinformatics.com.cn, accessed on 22 November 2023) and Cytoscape 3.8.0 software.

### 2.9. Detection of the Inflammatory Factor IL-6

The Vero cells were infected with TBEV (10TCID_50_) for 2 h, washed with PBS, and exposed to various concentrations (5, 10, and 20 µM) of MIN for 48 h. The values of IL-6 in the cell supernatant were quantitatively measured with a commercial ELISA kit (ml043278, Enzyme-linked Bio, Shanghai, China).

### 2.10. Statistical Analysis

GraphPad Prism 8.0 software was employed to generate graphical representations. Image J 1.8 software was utilized for the quantitative analysis of the Western blot images, focusing on the gray value of each target protein blot. The results of all data analyses are reported as the mean ± standard error of the mean (SEM). The difference between the two groups was analyzed using an independent sample *t*-test with a bilateral test at a significance level of α = 0.05. The significant differences were as follows: compared with the TBEV group, it was expressed by “*”, * (*p* < 0.05), ** (*p* < 0.01), and *** (*p* < 0.001), and compared with the control group, it was expressed by “#”, # (*p* < 0.05), ## (*p* < 0.01), and ### (*p* < 0.001). The data presented in this study were derived from a minimum of three independent trials.

## 3. Result

### 3.1. Evaluation of the Safety and Efficacy of Tetracycline Antibiotics in Vero Cells

To determine the suitable concentration of tetracycline antibiotics, a CCK8 assay was used to analyze the effect of different tetracycline antibiotic concentrations on the viability of Vero cells. The results showed that the survival ratios of Vero cells decreased with an increase in antibiotic concentration. No significant cytotoxicity was observed for MIN at concentrations less than 20 μM (Figure 1A), for TET and DOX at concentrations less than 10 μM (Figure 1B,C), and for AMT at concentrations less than 40 μM (Figure 1D). Thus, in subsequent experiments, we utilized less than the maximum safe concentrations of these drugs.

To assess the potential effect of tetracycline antibiotics on TBEV replication, the Vero cells were infected with TBEV (10 TCID_50_) for two hours, exposed to various concentrations of tetracycline antibiotics, and harvested after incubation for 48 h. Then, the total RNA of the Vero cells was extracted, and the transcriptional levels of TBEV RNA were comparatively assessed using RT-qPCR with TBEV RNA-specific primers. The results showed that 5, 10, and 20 μM MIN significantly inhibited TBEV replication in a dose-dependent manner. Additionally, TET and DOX also inhibited TBEV replication, but their inhibitory effects were significantly inferior to MIN (Figure 1E). Thus, MIN was selected to further explore the mechanism of TBEV replication inhibition. Amantadine was used as a positive control drug [20,21].

### 3.2. MIN Inhibits TBEV Replication in Vero Cells

Two different modes of administration were implemented to explore the impact of MIN on TBEV replication in TBEV-infected Vero cells: a preventive treatment (pre-treatment) and a post-treatment (Figure 2A). The RT-qPCR results showed that the average level of TBEV RNA in the TBEV group was about 0.5-fold lower than those in the 20 μM MIN and 10 μM AMT post-treatment groups, showing a significant difference. However, the average levels of TBEV RNA in the 20 μM MIN and 10 μM AMT pre-treatment groups were not statistically different compared to that of the TBEV group (Figure 2B).

In order to determine the optimal MOIs of TBEV infection with MIN in the Vero cells, we infected the cells with TBEV at MOIs of 10TCID_50_ and 100TCID_50_ for 2 h, followed by incubation with MIN and AMT for 48 h. The cells were harvested, and the TBEV RNA and protein levels were detected using RT-qPCR and IFA, respectively. The RT-qPCR results show that both 20 μM MIN and 10 μM AMT effectively inhibited TBEV RNA expression levels at different MOIs of TBEV infection, showing a significant difference (Figure 3A). IFA with the TBEV antibody illustrates darker green signals in the 20 μM MIN groups at different MOIs of TBEV infection, showing a notably lower intensity of fluorescence than that in the TBEV group (Figure 3B). However, The intensity of green fluorescence in the 10 μM AMT group was only weakened at 10TCID_50_ TBEV infection.

To further verify the inhibitory effect of MIN on TBEV replication, the Vero cells were infected with TBEV (10TCID_50_) for 2 h and exposed to various concentrations (5, 10, and 20 μM) of MIN and harvested after incubation at different times (24, 48, and 72 h). The results show that the gene, protein, and viral titer levels of TBEV increased with a prolongated incubation time (Figure 4). The NP mRNA (Figure 4A) and viral titers levels (Figure 4B) were significantly decreased at 24, 48, and 72 h after MIN treatment compared with those of the TBEV group. Further, the effect of MIN on TBEV protein synthesis was examined using IFA. Green fluorescence indicated the TBEV protein, while blue fluorescence showed nuclear staining. This revealed that MIN and AMT inhibited TBEV protein synthesis. Additionally, MIN showed a dose-dependent effect at all of the time points investigated (Figure 4C). 

### 3.3. Transcriptomic Analysis Results

The above experiments showed that MIN (20 μM) treatment had a better inhibitory effect on the TBEV-infected Vero cells. Therefore, in order to further elucidate the mechanism of MIN inhibiting TBEV replication, the DEGs between the TBEV-infected group and the MIN (20 µM) post-treatment 48-h group were determined using transcriptome analysis. The DEGs were selected with a *p* value < 0.05 and a |fold change| > 2. A total of 404 genes were identified in MIN vs. TBEV, of which 91 genes were up-regulated and 313 genes were down-regulated (Figure 5A). The transcriptome sequencing results of the DEGs with a significant *p* value and large fold change are shown in Figure 5B, of which the up-regulated genes were *RyR2*, *SNAP25*, *PRKCG*, and *FGF2*, and the down-regulated genes were *PLCB4*, *PDGFRA*, *SYT11*, *PLCB2*, and *CALML4*. In order to screen out the key targets, a protein-protein interaction (PPI) network of the 404 targets was constructed. The PPI results exhibit complex interactions between the targets, including 379 nodes and 228 edges (Figure 5C). The large areas and dark colors represent higher values for these genes, indicating a more significant effect of the MIN treatment on these genes in TBEV-infected Vero cells (Figure 5D). In addition, the top 15 core genes with high degree values are shown in Figure 5E. These target proteins exhibit significant interactions with other proteins and could potentially play a crucial role in the MIN treatment of TBEV infection.

In order to further investigate the relevant biological functions of the DEGs, GO and KEGG enrichment analyses were performed using the DAVID database. The biological processes (BP) enriched by the DEGs included the positive regulation of gene expression, the cellular response to mechanical stimulus, the positive regulation of ERK1 and ERK2 cascades, the negative regulation of interleukin-6 production, and others. The cellular components (CC) associated with the DEGs were mainly in the extracellular region. The molecular functions (MF) mainly included calcium ion binding, receptor binding. The terms from the GO enrichment analysis were visualized in a bubble plot, where the size of the bubbles corresponds to the number of genes and the color of the node represents the *p* value (Figure 6A).

Further, the main terms of GO enrichment analysis with their associated genes are shown in a chord diagram (Figure 6B). The KEGG pathway annotation showed that the signal transduction subgroup had the largest number of annotated genes (Figure 6C). The Sankey-Bubble plot revealed the three most significantly enriched pathways, including pathways in cancer, the calcium signaling pathway, and the MAPK signaling pathway. The pathways in cancer include the MAPK signaling pathway. Combined with gene enrichment on the left, KEGG pathway enrichment analysis showed that the most enriched genes were *PRKCG*, *PDGFRA*, *FGF2*, *CALML4*, *PLCB2* and *RYR2* (Figure 6D). The network model of interactions between the signaling pathways revealed that the calcium signaling pathway, the Rap1 signaling pathway, the Ras signaling pathway, and the MAPK signaling pathway were interconnected with many other signaling pathways, highlighting their significance in signal transduction (Figure 6E).

Taken together, GO and KEGG enrichment analyses indicated that MIN may inhibit TBEV replication by regulating calcium channels and the MAPK-ERK signaling pathway and inhibiting the production of the inflammatory factor IL-6.

### 3.4. MIN Regulates Calcium Signaling Pathways in TBEV-Infected Vero Cells

Previous studies have revealed that calcium ions can affect the replication of *Orthoflavivirus* [22,23]. To verify the transcriptomic analysis results and investigate whether MIN could potentially impact calcium homeostasis in TBEV-infected Vero cells by regulating the calcium signaling pathway, some key genes associated with the calcium channels identified via transcriptome analysis were evaluated with individual specific primers, such as *CALML4*, *RYR2*, and *SNAP25*. The RT-qPCR results were consistent with the transcriptomic analysis results, confirming the accuracy of the transcriptome sequencing. Compared with the Vero cells group (control group), the mRNA expression of CALML4 and TBEV RNA significantly increased in the TBEV-infected group, while the changes were reversed in the 20 μM MIN and 10 μM AMT groups, with statistical significance (Figure 7A,B). Additionally, compared with the Vero cells group, the mRNA expression of RYR2 and SNAP25 decreased in the TBEV-infected group, while the 20 μM MIN treatment significantly increased the mRNA expression levels of both RYR2 and SNAP25 compared with those of the TBEV-infected group (Figure 7C). Meanwhile, results showed that, compared with the Vero cells group, the mRNA expression levels of *CALML4*, *RYR2*, and *SNAP25* were no different in the 20 µM MIN group. This suggests that the treatment with MIN in TBEV-infected Vero cells efficiently regulates the calcium signaling pathway-associated proteins.

### 3.5. MIN Inhibits the MAPK-ERK Signaling Pathway in TBEV-Infected Vero Cells

Tetracycline antibiotics can activate the MAPK-ERK signaling pathway. As a tetracycline antibiotic, MIN is capable of regulating the MAPK-ERK signaling pathway [24]. Meanwhile, the results of GO and KEGG enrichment analyses confirmed that the MAPK-ERK signaling pathway may play a crucial role in the MIN treatment of TBEV-infected Vero cells. To investigate the possible alteration of the MAPK-ERK signaling pathway in TBEV-infected Vero cells after exposure to MIN, some upstream genes of the MAPK-ERK signaling pathway were evaluated with individual-specific primers, including *FGF2*, *PDGFRA*, and *PLCB2*. Compared with the Vero cells group, the mRNA expression levels of *FGF2*, *PDGFRA*, and *PLCB2* were no different in the 20 µM MIN group. The relative mRNA value of *FGF2* was significantly increased, and those of *PDGFRA* and *PLCB2* were significantly decreased in the MIN-treated TBEV-infected Vero cells compared to those of the untreated cells (Figure 8A,B). The phosphorylation of ERK (p-ERK) is a marker of MAPK-ERK pathway activation. The Western blot assay results show that, compared with the Vero cells group, p-ERK/ERK were no different in the 20 µM MIN group (Appendix A (Appendix A include the raw Western blot data see Appendix A)). There was a dose-dependent decrease in the protein expression level of TBEV and p-ERK in the MIN-treated TBEV-infected Vero cells, and the effect was better than that of AMT (Figure 8C,D (Appendix A include the raw Western blot data)). These results indicate that treatment with MIN in TBEV-infected Vero cells efficiently inhibits the MAPK-ERK signaling pathway.

### 3.6. MIN Inhibits the Expression of Inflammatory Factor IL-6 in TBEV-Infected Vero Cells

IL-6 is an important factor in inflammatory responses. A previous study has identified the increased expression and secretion of IL-6 in TBE, which may mediate the cytokine storm [25]. Meanwhile, combined with the IL-6 expression-related gene *SYT11* and the GO term of the negative regulation of interleukin-6 production (GO:0032715), we explored whether MIN could inhibit the inflammatory response in TBEV-infected Vero cells by reducing the expression and secretion of IL-6. Compared with the control group, there was no difference in SYT11 mRNA expression in the 20µM MIN group, while the expressions of SYT11 and IL-6 were significantly increased in the TBEV-infected group, and this change was reversed by the MIN treatment, which was more effective than that of AMT (Figure 9A,B). The RT-qPCR results were consistent with the transcriptomic sequencing results. Furthermore, the level of IL-6 in the supernatants of the TBEV-infected Vero cells were measured using an ELISA kit assay, indicating a significant increase. However, the level of IL-6 in the supernatant of the TBEV-infected Vero cells decreased in a dose-dependent manner after the MIN treatment, and its effect was better than that of AMT (Figure 9C). In addition, results showed that a significant decrease in mRNA expression levels of IL-6 in the 20 µM MIN group compared with the Vero cells group, and a slight reduction in IL-6 levels in the supernatants. These data suggest that treatment of TBEV-infected Vero cells with MIN inhibits IL-6 expression and alleviates the inflammatory response.

## 4. Discussion

In the present study, we have investigated the effect of MIN on the inhibition of TBEV and cellular protection in TBEV-infected Vero cells. Our data indicate that post-treatment with MIN for 48 h effectively inhibits the expression of TBEV RNA and proteins and reduces the progeny virus titer in a dose-dependent manner within the safe concentration range of the drug. In addition, MIN also showed a good inhibitory effect on a 100-fold TCID_50_ TBEV infection. The transcriptome analysis results showed that the inhibition of TBEV replication by MIN could be related to the regulation of the calcium signaling pathway, the MAPK-ERK signaling pathway, and the expression of inflammatory factors.

Currently, there is no specific clinical treatment for patients with TBE, but there is a symptomatic treatment [26]. Consequently, there is an urgent need for the screening or development of novel anti-TBEV drugs. The re-application of clinically approved drugs could offer the quickest pathway to the development of new drugs. MIN, a second-generation semi-synthetic tetracycline antibiotic, is commonly utilized in clinical settings due to its potent antibacterial properties, high bioavailability, and ability to penetrate the blood-brain barrier [27]. Recent research has also explored the anti-orthoflavivirus effects of MIN in addition to its antimicrobial properties [28,29,30]. Here, our data show that treatment with a relatively low dose of MIN not only reduces the intracellular TBEV levels but also regulates the calcium and MAPK-ERK signaling pathways and suppresses the expression of the inflammatory factor IL-6 in TBEV-infected Vero cells. Furthermore, by targeting the MAPK-ERK signaling pathway, MIN can stimulate the expression of antiviral genes, leading to effective suppression of DENV replication [17].

An imbalance of calcium homeostasis has been repeatedly reported in *Orthoflavivirus*-related diseases, mainly due to promotion of viral replication and the maintenance of infection [31]. Excessive calcium will induce intracellular enzyme cascades and inflammatory responses, and subsequently cause cell damage [32,33]. In order to maintain intracellular calcium homeostasis, the calcium receptors and channels play an important role. Calmodulin-like 4 (CALML4) is a subunit of calcium receptors in cells, which affects cell biological processes, such as signal transduction, protein phosphorylation, and gene expression regulation, by binding calcium ions and regulating calmodulin kinases [34]. Ryanodine receptor 2 (RYR2) and Synaptosome-associated protein 25 (SNAP25) are associated with the calcium channels. RYR2 is a major endoplasmic reticulum calcium channel and can regulate calcium ion flow between the endoplasmic reticulum and cytoplasm [35]. SNAP25 is a presynaptic membrane protein that negatively regulates voltage-gated calcium channels [36]. Our study showed that the MIN post-treatment was able to regulate the levels of CALML4, RYR2, and SNAP25, thereby restoring the expression of calcium signaling pathway-associated proteins in the TBEV-infected Vero cells. In fact, MIN has been found to be a calcium chelator that is capable of restoring rotenone-induced calcium ion deregulation, thereby the primary cortical neurons are protected from calcium ions [37]. Although there is no direct evidence to ascribe the MIN-mediated reduction in TBEV to the restoration of calcium homeostasis, further studies to explore the association between these two phenomena are warranted.

Previous studies have shown that MAPK-ERK signaling plays a key role in *Orthoflavivirus* replication and release [38,39]. Extracellular growth factors and receptor tyrosine kinases are important factors that activate the MAPK-ERK signaling pathway [40,41]. Fibroblast growth factor 2 (FGF2) is a type of growth factor, and Platelet-Derived Growth Factor Receptor α (PDGFRA) is a member of the receptor tyrosine kinase family. Some studies have shown that FGF2 can negatively regulate the expression level of PDGFRA [42]. PDGFRA can activate Phosphatidylinositol-4,5-bisphosphate (PIP_2_) by binding to ligand PDGF. PLCB2, a phosphodiesterase, catalyzes the hydrolysis of PIP_2_ to inositol 1,4,5-trisphosphate (IP_3_) and diacylglycerol (DAG). DAG can activate PKC, thereby regulating the MAPK-ERK signaling pathway [43,44]. Similar to a previous study showed that MIN affects DENV replication by inhibiting MAPK-ERK [17], our current study also found that MIN can inhibit the MAPK-ERK signaling pathway in TBEV-infected Vero cells, as indicated by the increased levels of upstream FGF2 and the decreased levels of PDGFRA and PLCB2, thereby inhibiting the expression level of p-ERK. In addition, IP_3_ binds to the calcium channel receptor IP_3_R on the endoplasmic reticulum and promotes the release of calcium ions, which can bind to CaM kinase II (CaMKII) or activate protein kinase C (PKC), thereby activating the MAPK-ERK signaling pathways. This suggests that MAPK-ERK signaling can also be triggered by calcium ions, which may highlight the cross-linking of the MAPK-ERK and calcium signaling pathways mediated by MIN in TBEV-infected Vero cells.

IL-6, a multifunctional cytokine regulated by the MAPK-ERK signaling pathway, plays a crucial role in the body’s defense against inflammation [45,46]. Synaptotagmin 11 (SYT11) is a synaptotagmin that inhibits the secretion of the cytokine IL-6 [47]. Other studies have shown that TBEV infection produces large amounts of IL-6, which triggers an inflammatory response [48]. Reducing the increased IL-6 and SYT11 levels in TBEV-infected Vero cells using MIN may reflect the reversal of inflammatory responses by TBEV. As it is a tetracycline antibiotic, MIN mediates many biological processes in TBEV-infected Vero cells.

Our data demonstrated that the post-treatment of TBEV-infected Vero cells with MIN for 48 h can regulate calcium signaling pathway-associated proteins, reduce the level of TBEV, inhibit the activation of the MAPK-ERK signaling pathways, reduce the expression of the inflammatory factor IL-6, and thus alleviate the intracellular inflammatory response (Figure 10). Therefore, the protective effect of MIN on TBEV-infected cells in vitro makes it worthy of further investigation. Currently, there is no established treatment regimen for patients with TBE, underscoring the importance of developing a reliable anti-TBEV compound to expedite drug development. This study demonstrated the inhibitory effects of MIN on TBEV replication in vitro and provided initial insights into its mechanism of action, and using MIN to inhibit TBEV infectivity and improve patients’ clinical progression in vivo is anticipated.

## Figures and Tables

**Figure 1 viruses-16-01055-f001:**
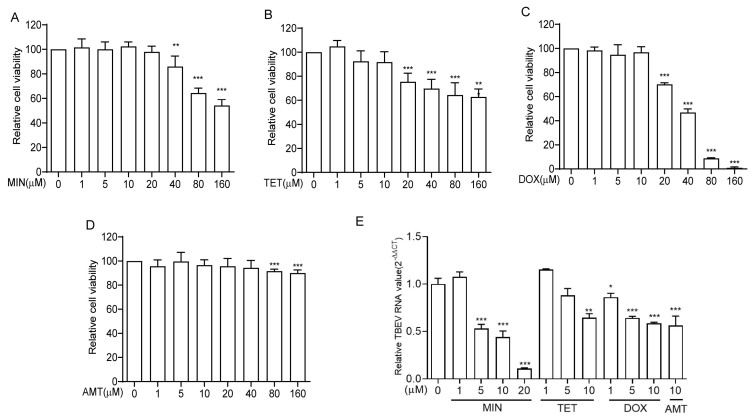
Evaluation of the influence of tetracycline antibiotics on the cell viability and inhibition of TBEV replication in Vero cells. (**A**–**D**) The effects of tetracycline antibiotics on cell viability in the Vero cells. The Vero cells were incubated with MIN, TET, DOX, and AMT for 72 h. (**E**) The effects of tetracycline antibiotics on the replication of TBEV was assessed using RT-qPCR. The dosages of the tetracycline antibiotics are shown below each graph. The difference between the two groups was analyzed using an independent sample *t*-test (***, ** and * indicate *p* < 0.001, *p* < 0.01 and *p* < 0.05, respectively).

**Figure 2 viruses-16-01055-f002:**
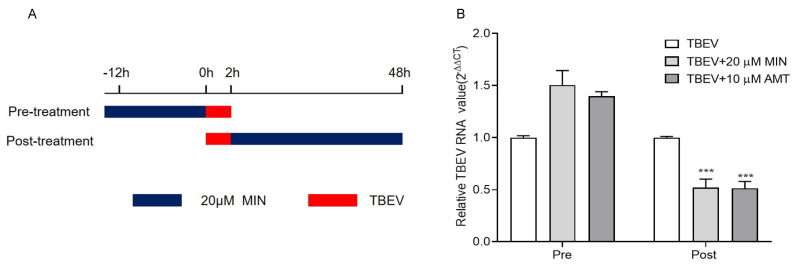
Evaluation of influence of MIN on the inhibition of TBEV replication under different modes of administration in Vero cells. (**A**) Diagram of different modes of administration, Blue reresents the duration of treatment and red represents the infection time of TBEV. (**B**) Comparative analysis of different modes of administration using RT-qPCR. The difference between the two groups was analyzed using an independent sample *t*-test (*** indicate *p* < 0.001).

**Figure 3 viruses-16-01055-f003:**
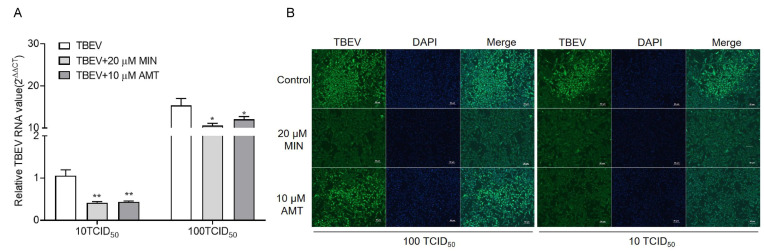
Evaluation of the influence of MIN on the inhibition of TBEV replication under different MOIs in the Vero cells. The effect of MIN on TBEV replication in different MOIs was detected using RT-qPCR (**A**) and IFA (**B**). The difference between the two groups was analyzed using an independent sample *t*-test (** and * indicate *p* < 0.01 and *p* < 0.05, respectively).

**Figure 4 viruses-16-01055-f004:**
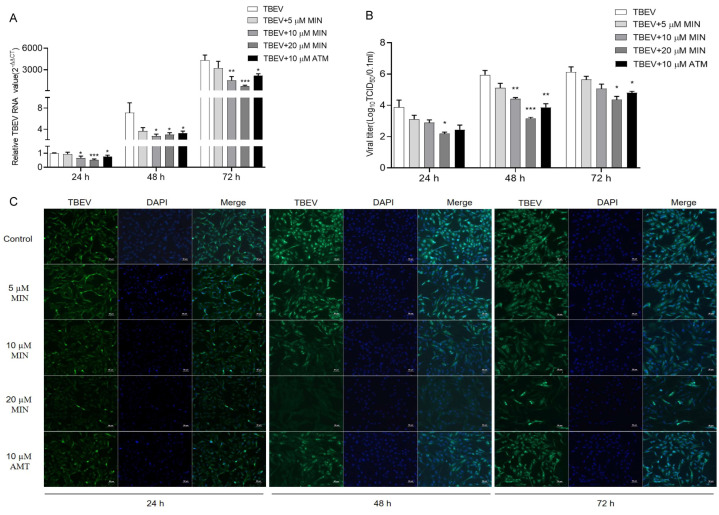
Evaluation of the influence of MIN on the inhibition of TBEV replication in the Vero cells. The optimal effective concentration and time of MIN inhibiting TBEV replication in the Vero cells were determined using RT-qPCR (**A**), viral titers (**B**), and IFA (**C**). The dosages of MIN are shown above each graph. The difference between the two groups was analyzed using an independent sample *t*-test (***, ** and * indicate *p* < 0.001, *p* < 0.01 and *p* < 0.05, respectively).

**Figure 5 viruses-16-01055-f005:**
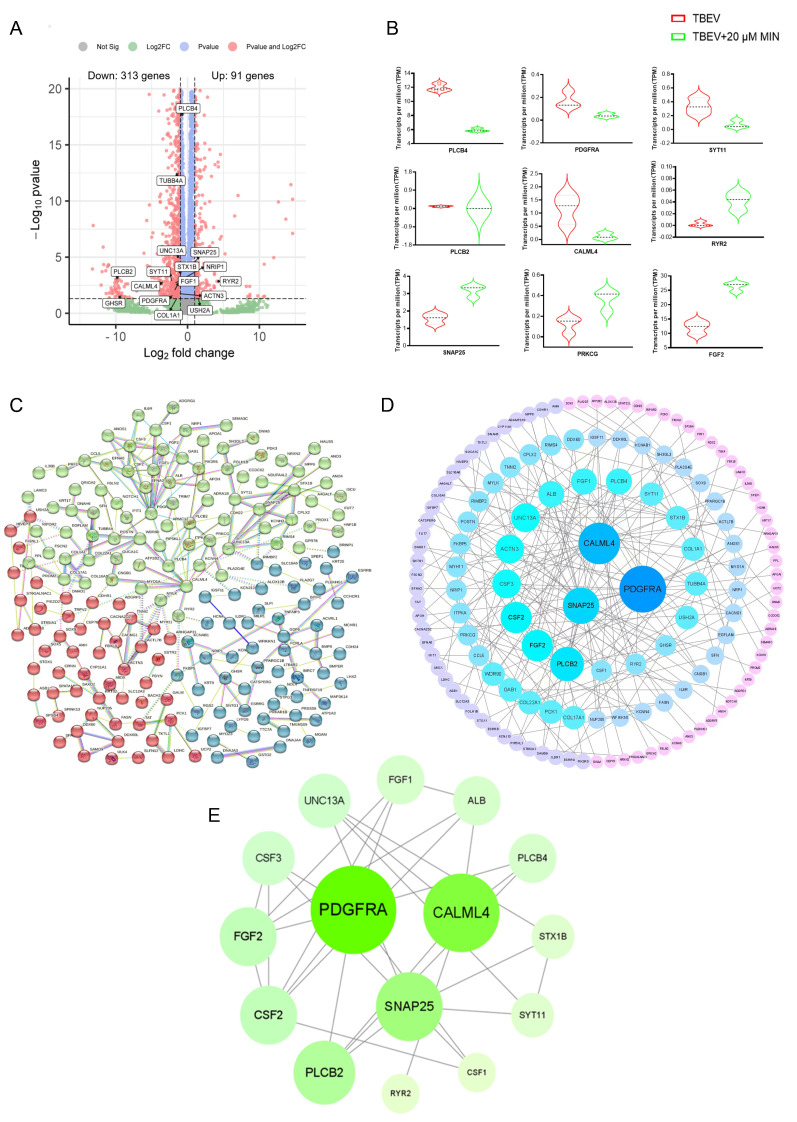
Gene expression profile under TBEV infection condition with MIN treatment. (**A**) Volcano plot of DEGs. (**B**) Transcriptomic sequencing results of *PRKCG*, *PDGFRA*, *FGF2*, *CALML4*, *PLCB2*, *PLCB4*, *SYT11*, *SNAP25,* and *RYR2*. (**C**) PPI network of DEGs. (**D**) Circle diagram of DEGs ranked by degree. (**E**) PPI network of 15 core genes.

**Figure 6 viruses-16-01055-f006:**
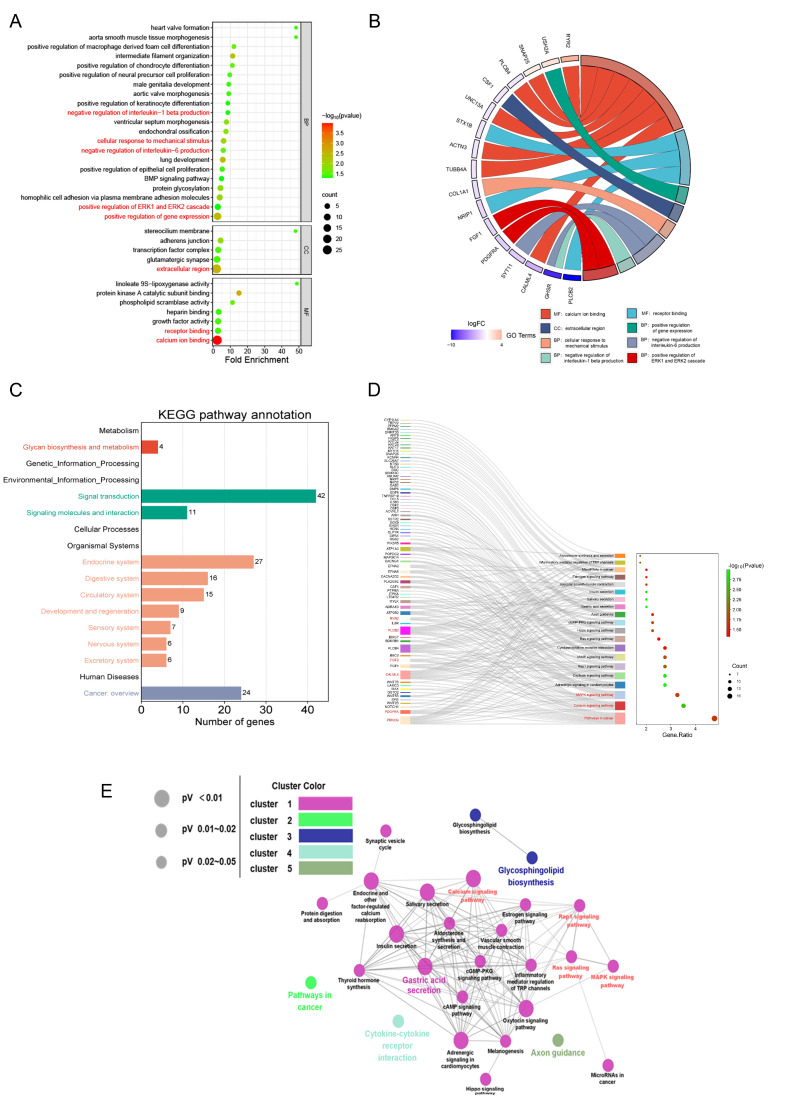
Transcriptional profiles of TBEV infection under MIN treatment. (**A**) The significantly enriched GO terms for DEGs. (**B**) A chord diagram between specific GO terms and relevant DEGs. (**C**) The results of KEGG pathway annotation. (**D**) A Sankey-Bubble plot of significant KEGG pathways and relevant DEGs. (**E**) A network model of interactions between significant KEGG pathways.

**Figure 7 viruses-16-01055-f007:**
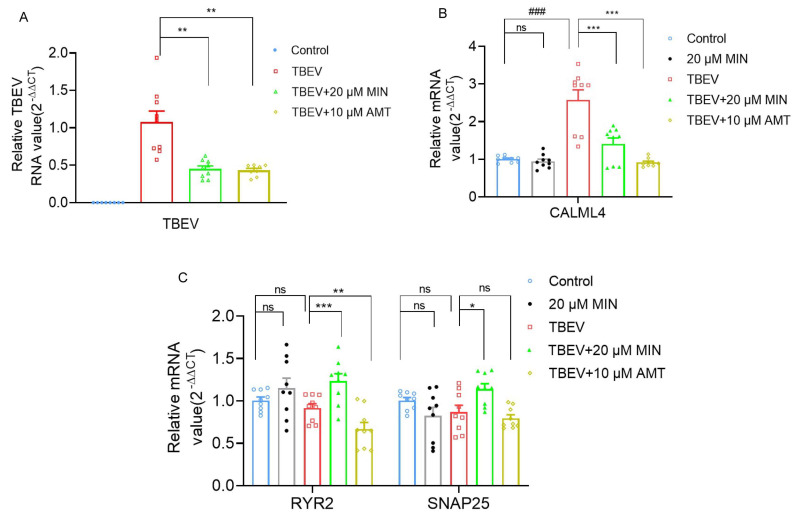
Evaluation of the effects of MIN on regulating calcium signaling pathway in TBEV-infected Vero cells. (**A**) RT-qPCR of RNA levels of TBEV RNA. (**B**,**C**) RT-qPCR of mRNA levels of *CALML4 RYR2,* and *SNAP25*. The difference between the two groups was analyzed using an independent sample *t*-test, compared with the TBEV group, it was expressed by “*”, * (*p* < 0.05), ** (*p* < 0.01), and *** (*p* < 0.001), and compared with the Vero cells group, it was expressed by “#”, ### (*p* < 0.001), the ns symbol represented no significant difference.

**Figure 8 viruses-16-01055-f008:**
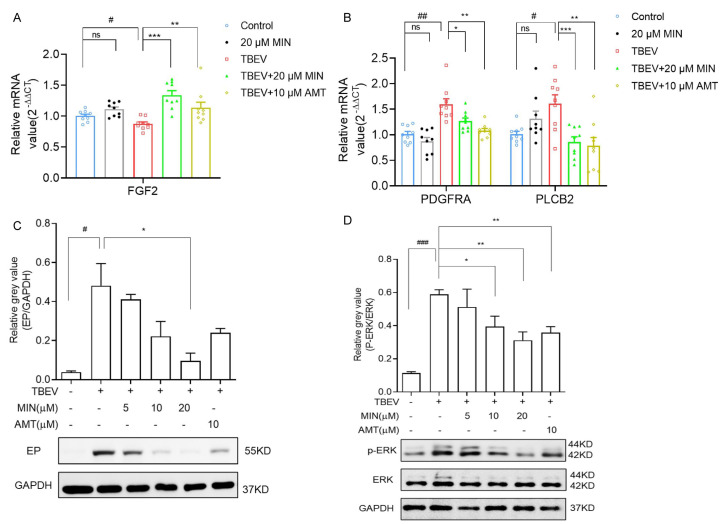
Evaluation of the effects of MIN on inhibition of the MAPK-ERK signaling pathway in TBEV-infected Vero cells. (**A**,**B**) The RT-qPCR of the mRNA level of *FGF2*, *PDGFRA,* and *PLCB2*. (**C**,**D**) Western blot of the expression levels of the TBEV envelope protein (EP) and p-ERK/ERK. Quantitative analysis of the average gray values of TBEV and p-ERK/ERK after normalization against GAPDH. The difference between the two groups was analyzed using an independent sample *t*-test, compared with the TBEV group, it was expressed by “*”, * (*p* < 0.05), ** (*p* < 0.01), and *** (*p* < 0.001), and compared with the Vero cells group, it was expressed by “#”, # (*p* < 0.05), ## (*p* < 0.01), and ### (*p* < 0.001), the ns symbol represented no significant difference.

**Figure 9 viruses-16-01055-f009:**
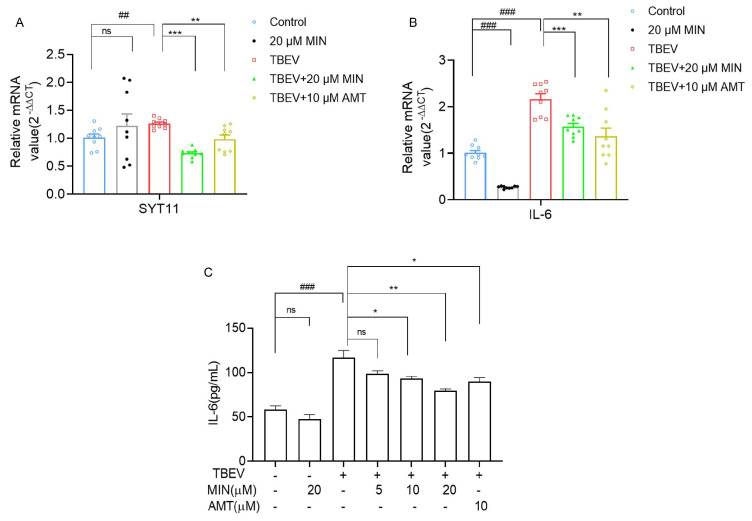
Evaluation of the effects of MIN on the reduced IL-6 expression level in TBEV-infected Vero cells. (**A**,**B**) RT-qPCR off the mRNA level of SYT11 and IL-6. (**C**) The level of IL-6 in the supernatant of the TBEV-infected Vero cells measured using an ELISA kit. Control: without TBEV infection. The difference between the two groups was analyzed using an independent sample *t*-test, compared with the TBEV group, it was expressed by “*”, * (*p* < 0.05), ** (*p* < 0.01), and *** (*p* < 0.001), and compared with the Vero cells group, it was expressed by “#”, ## (*p* < 0.01), and ### (*p* < 0.001); the ns symbol represented no significant difference.

**Figure 10 viruses-16-01055-f010:**
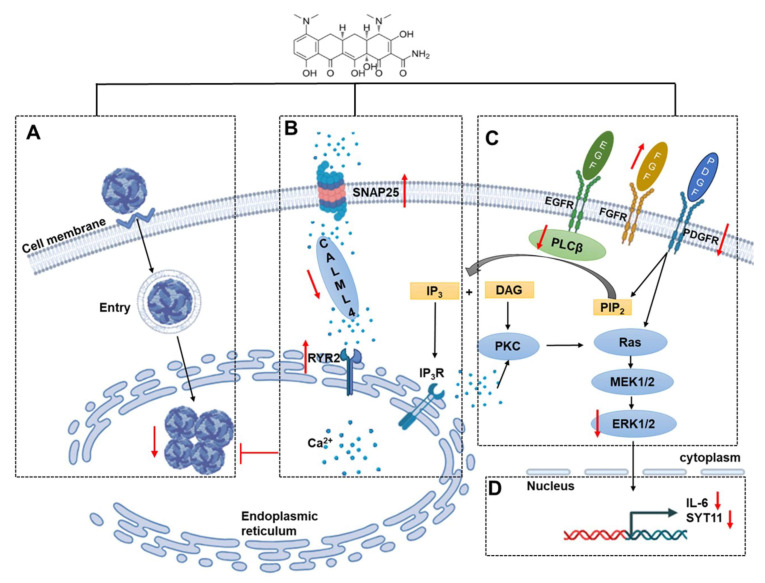
A hypothetical schema of the protective role of MIN in TBEV-infected Vero cells. (**A**) MIN directly reduces the intracellular TBEV levels. (**B**) MIN regulates the calcium receptors and calcium channel, calmodulin, and inhibits TBEV replication. (**C**) MIN inhibits the expression of the MAPK-ERK signaling pathway by up-regulating FGF2 and down-regulating PDGF expression. (**D**) MIN alleviates the inflammatory response by reducing IL-6 and SYT11 expression.

## Data Availability

Data will be made available upon request.

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
