# Peer review of "Minocycline Inhibits Tick-Borne Encephalitis Virus and Protects Infected Cells via Multiple Pathways"

_viruses, 2024, doi:10.3390/v16071055_

Round 1

Reviewer 1 Report

Comments and Suggestions for Authors

The manuscript is devoted to studying the mechanism of the inhibitory activity of the antibiotic minocycline in vitro. It is extremely important to identify the interaction of various signaling pathways. In addition, this may be the next step in obtaining a therapeutic drug for TBE. This work has additional practical relevance due to the fact that a tick bite can cause a mixed virus-bacterial infection and you need to know how an antibiotic can affect the course of a viral infection.

However, there are many inaccuracies in the work. Therefore, the manuscript cannot be published in this form. 1. The nomenclature of viruses has changed. Now, according to ICTV, the genus Flavivirus (italics) is called Orthoflavivirus (italics). The title of the entire text will need to be changed. 2.In the text, the authors write about viral RNA as a nucleocapsid. A nucleocapsid is a nucleic acid associated with a viral capsid protein. How the viral capsid is structured in orthoflaviviruses is still unclear. For Flavivivrus there is no concept of mRNA, since viral RNA is the template for protein synthesis, i.e. mRNA. Inside cells infected with orthoflaviviruses, viral RNA can be associated with the nucleocapsid protein, but mostly it is either contained in the replication complex or directly in the virion. Accordingly, the authors should remove the word nucleocapsid everywhere in the text and replace it with viral RNA, and also correct Figure 10. 3. 20-21 line. MIN cannot influence multiplicity of infection. Also, poor English. 3. paragraph 2.1. The name of the strain and its passage history of the virus should be indicated so that it is clear how much its sequence may differ from that given in GenBank. For dengue virus, it has been shown that tetricyclics can interact with surface glycoproteins and thus inhibit virus reproduction. In this case, the sequence is important. 5. Ibid. Why was the viral suspension filtered through a filter? 6. 81 lines. "The level of TBEV" should be written "The level of TBEV reproduction". 7. It is worth explaining somewhere why Amantadine salt is used as a positive control. 8. Paragraph 2.4. The authors performed freeze-thaw tests before quantifying the virus. This is somewhat incorrect. Only extracellular virions of orthoflaviviruses are known to be infectious. When the amount of viral RNA in preparations is measured after cell destruction, the amount of both infectious and non-infectious virus is assessed. When titrated by CPE, the number of infectious virions is determined, but non-infectious intracellular virions released from destroyed cells can reduce titers because they can interact with receptors on the surface of cells. Since the authors see a decrease in the amount of both viral RNA and infectious virus, freezing-thawing will not affect the conclusions, but this fact needs to be discussed. 9. 118 line. It is necessary to indicate to which strain the hyperimmune serum was obtained or its number, if it is commercial. 10. Paragraph 2.8. It should be indicated how many replicates were used for transcriptomic analysis. 10. Paragraph 2.10. Are the authors sure they can use t-test? 11. 210-211 line. The authors write that the pre-treatment increased slightly, but the post-treatment increased significantly, although the change was the same. The phrase needs to be corrected. 12. 269-275 lines. The authors introduce unnecessary abbreviations that make the text difficult to read. 13. TheFig. it shows bars that are statistically significantly different..., but it is not written which bar they differ from. It is not specified what ### means 14. Fig. S1. The reproduction rate is not an infective effect. Bad English.  

Author Response

Dear Editors and Reviewers:

Thank you for your letter and for the reviewers, comments concerning our manuscript entitled “Minocycline Inhibits Tick-Borne Encephalitis Virus and Protects Infected Cells via Multiple Pathways”(ID: viruses-3015788). Those comments are all valuable and very helpful for revising and improving our paper, as well as the important guiding significance to our researches. We have studied comments carefully and have made correction which we hope meet with approval. Revised portion are marked in red in the manscript. The main corrections in the paper and the responds to the reviewer's comments areas flowing:

Comments 1: The nomenclature of viruses has changed. Now, according to ICTV, the genus Flavivirus (italics) is called Orthoflavivirus (italics). The title of the entire text will need to be changed.

Response 1: Thank you for your professional suggestions. We have corrected the "genus flavivirus" to "orthoflavivirus" in the manuscript and marked it in red font. Relative information has been modified in lines 35, 40, 57, 304, 389, 396 and 416.

Comments 2: In the text, the authors write about viral RNA as a nucleocapsid. A nucleocapsid is a nucleic acid associated with a viral capsid protein. How the viral capsid is structured in orthoflaviviruses is still unclear. For Flavivivrus there is no concept of mRNA, since viral RNA is the template for protein synthesis, i.e. mRNA. Inside cells infected with orthoflaviviruses, viral RNA can be associated with the nucleocapsid protein, but mostly it is either contained in the replication complex or directly in the virion. Accordingly, the authors should remove the word nucleocapsid everywhere in the text and replace it with viral RNA, and also correct Figure 10.

Response 2: Thank you for your professional suggestions. We have corrected the" TBEV nucleocapsid mRNA "to "TBEV RNA". Relative information has been modified in lines 194-195, 211, 213, 222, 224, 311, 323 and 377. And we have corrected Figure 10.

Comments 3: 20-21 line. MIN cannot influence multiplicity of infection. Also, poor English.

Response 3: Thank you for your professional suggestions. The manuscript has been edited by the recommended editing services. We have corrected it to "The inhibitory effect of MIN on different TBEV multiplicities of infection (MOIs) in vero cells." and marked it in red font. Relative information has been modified in lines 20-21.

Comments 4: paragraph 2.1. The name of the strain and its passage history of the virus should be indicated so that it is clear how much its sequence may differ from that given in GenBank. For dengue virus, it has been shown that tetricyclics can interact with surface glycoproteins and thus inhibit virus reproduction. In this case, the sequence is important.

Response 4: Thank you for your professional suggestions. We conducted a sequence comparison between the sequenced TBEV genome and the sequence given in GenBank, revealing differences in several genes at the 5' and 3' ends, while genes at other positions were the same. Consequently, we posit that no novel surface glycoproteins are on the viral envelope.Relative information has been modified in lines 77-78.

Comments 5:. Ibid. Why was the viral suspension filtered through a filter?

Response 5: In the text, we employed a combination of centrifuge centrifugation and filter filtration to eliminate cellular debris from the viral suspension, in order to obtain a pure virus suspension and ensure no interference with subsequent experiments.

Comments 6: 81 lines. "The level of TBEV" should be written "The level of TBEV reproduction".

Response 6: Thank you for your professional suggestions. We have corrected "The level of TBEV" to "The level of TBEV reproduction" and marked it in red font. Relative information has been modified in line 83.

Comments 7: It is worth explaining somewhere why Amantadine salt is used as a positive control.

Response 7: In the text, we employed Amantadine salt as a positive control. There are two reasons. Firstly, Studies have demonstrated that amantadine can inhibit a range of viruses, including the dengue virus from the orthofavivirus genus[1-2]. Furthermore, amantadine can penetrate the blood-brain barrier[3-4], which is consistent with the main study object minocycline in this study. It also lays the groundwork for further research on the effect of intracranial treatment. Therefore, amantadine was selected as the positive control drug. Relative information has been modified in lines 199-200.

[1]Koff WC, Elm JL, Halstead SB.1980.Inhibition of dengue virus replication by amantadine hydrochloride. Antimicrob Agents Chemother18:.https://doi.org/10.1128/aac.18.1.125IF: 4.9 Q1 IF: 4.9 Q1

[2]Lin CC, Chen WC. Treatment Effectiveness of Amantadine Against Dengue Virus Infection. Am J Case Rep. 2016 Dec 5;17:921-924. doi: 10.12659/AJCR.901014. PMID: 27920420; PMCID: PMC5158130.

[3] Spector R. Transport of amantadine and rimantadine through the blood-brain barrier. J Pharmacol Exp Ther. 1988 Feb;244(2):516-9. PMID: 3346834.

[4] Suzuki T, Aoyama T, Suzuki N, Kobayashi M, Fukami T, Matsumoto Y, Tomono K. Involvement of a proton-coupled organic cation antiporter in the blood-brain barrier transport of amantadine. Biopharm Drug Dispos. 2016 Sep;37(6):323-35. doi: 10.1002/bdd.2014IF: 2.1 Q4 . Epub 2016 Aug 24. Erratum in: Biopharm Drug Dispos. 2016 Nov;37(8):507. PMID: 27146715.

Comments 8:. Paragraph 2.4. The authors performed freeze-thaw tests before quantifying the virus. This is somewhat incorrect. Only extracellular virions of orthoflaviviruses are known to be infectious. When the amount of viral RNA in preparations is measured after cell destruction, the amount of both infectious and non-infectious virus is assessed. When titrated by CPE, the number of infectious virions is determined, but non-infectious intracellular virions released from destroyed cells can reduce titers because they can interact with receptors on the surface of cells. Since the authors see a decrease in the amount of both viral RNA and infectious virus, freezing-thawing will not affect the conclusions, but this fact needs to be discussed.

Response 8: Thank you for your professional suggestions. We agree with this comment. Therefore, to eliminate the effects of non-infectious intracellular virions, we redesigned the experiment, the collected supernatant was diluted directly in gradient and then infected cells. The new experimental results were shown in Figure 4B. Relative information has been modified in lines 105-107.

Comments 9:. 118 line. It is necessary to indicate to which strain the hyperimmune serum was obtained or its number, if it is commercial.

Response 9: Thank you for your professional suggestions. In the text, the TBEV-specific antibody employed in our research is our laboratory provided and not commercial. Relative information has been modified in line 119.

Comments 10:. Paragraph 2.8. It should be indicated how many replicates were used for transcriptomic analysis.

Response 10: Thank you for your professional suggestions. We agree with this comment. Therefore, we added the number of groups and replicates and marked it in red font. Relative information has been modified in lines 152-153.

Comments 11:. Paragraph 2.10. Are the authors sure they can use t-test?

Response 11: Thank you for your professional suggestions. The selected data are normally distributed, so t-test can be applied to compare the differences between the two groups. For greater clarity, we elaborate in 2.10. and marked it in red font. Relative information has been modified in lines 174-180.

Comments 12:. 210-211 line. The authors write that the pre-treatment increased slightly, but the post-treatment increased significantly, although the change was the same. The phrase needs to be corrected.

Response 12: Thank you for your professional suggestions. We have corrected to " However, the average levels of TBEV RNA in the 20 μM MIN and 10 μM AMT pre-treatment groups were not statistically different compared to that of the TBEV group " and marked it in red font. Relative information has been modified in lines 212-214.

Comments 13:. 269-275 lines. The authors introduce unnecessary abbreviations that make the text difficult to read.

Response 13: Thank you for your professional suggestions.We adjusted the content and the abbreviations used in this article have been explained in lines 461-466. We are very sorry for the inconvenience caused to your reading.

Comments 14:. TheFig. it shows bars that are statistically significantly different..., but it is not written which bar they differ from. It is not specified what ### means

Response 14: Thank you for your professional suggestions. In 2.10 Statistical analysis, we made a detailed supplement of data significance and marked it in red font. Relative information has been modified in lines 177-179.

Comments 15:. Fig. S1. The reproduction rate is not an infective effect. Bad English.

Response 15: Thank you for your professional suggestions. Therefore, we have corrected " The infective effect of TBEV for different cells" to " The level of TBEV reproduction in different cells" and marked it in red font.

Reviewer 2 Report

Comments and Suggestions for Authors

Dear Authors,

I would like to congratulate you on preparing an interesting manuscript. However, upon reviewing the manuscript, I have identified specific areas that require corrections and explanations.

The manuscript requires some minor grammar and style corrections.

Lines 45-47: “Although TBEV and TBE has been known for decades, there are currently still no effective therapeutic drugs and preventive vaccines for TBE.” I must disagree; vaccines against TBEV are commonly available in Europe and Russia. The first vaccine was developed in 1937 in the former Soviet Union and was derived from mouse brain but resulted in frequent adverse effects [1]. Later, in the early 1970s, the first tissue culture-derived TBEV vaccine was developed [2]. Currently, there are several widely used vaccines (Encepur N, FSME-Immun CC and Ticovac, Encevir-Neo, and  Klesh-E-Vak) [1,2]. Hence, vaccination remains an effective way to prevent TBE.

Line 142: “Real-time quantitative PCR (RT-qPCR)”. Please pay attention to using the proper names of molecular techniques. The “RT-PCR” should only be used to describe reverse transcription PCR and not the real-time PCR [Bustin, 2009]. Herein, since RNA was your matrix, reverse transcription was used to generate cDNA, and further quantitative amplification was applied; the proper name should be quantitative reverse transcription polymerase chain reaction (RT-qPCR or qRT-PCR) or real-time reverse transcription PCR (abbrev also RT-qPCR or qRT-PCR). Interestingly, herein, you have used both two-step and one-step approaches to RT-qPCR.

Unfortunately, minocycline's antiviral potential was significantly dependent on the infectious dose of TBEV, as shown in Figure 3A (10-fold vs 100-fold TCID50). This is an important limitation and should be underlined and commented on more in the discussion.

References

1. World Health Organization. (2011). Vaccines against tick-borne encephalitis : WHO position paper = Note de synthèse : position de l'OMS sur les vaccins contre l'encéphalite à tiques. Weekly Epidemiological Record = Relevé épidémiologique hebdomadaire, 86 (24), 241 - 256. https://iris.who.int/handle/10665/241769

2. Barrett, P. N., Schober-Bendixen, S., & Ehrlich, H. J. (2003). History of TBE vaccines. Vaccine, 21 Suppl 1, S41–S49. https://doi.org/10.1016/s0264-410x(02)00814-9

3. Bustin, S. A., Benes, V., Garson, J. A., Hellemans, J., Huggett, J., Kubista, M., Mueller, R., Nolan, T., Pfaffl, M. W., Shipley, G. L., Vandesompele, J., & Wittwer, C. T. (2009). The MIQE guidelines: minimum information for publication of quantitative real-time PCR experiments. Clin Chem, 55(4), 611-622. https://doi.org/10.1373/clinchem.2008.112797

Comments on the Quality of English Language

The text would benefit from some minor language and style revisions to improve its clarity and coherence.

Author Response

Dear Editors and Reviewers:

Thank you for your letter and for the reviewers, comments concerning our manuscript entitled “Minocycline Inhibits Tick-Borne Encephalitis Virus and Protects Infected Cells via Multiple Pathways”(ID: viruses-3015788). Those comments are all valuable and very helpful for revising and improving our paper, as well as the important guiding significance to our researches. We have studied comments carefully and have made correction which we hope meet with approval. Revised portion are marked in red in the manuscript. The main corrections in the paper and the responds to the reviewer's comments areas flowing:

Comments 1: Lines 45-47: “Although TBEV and TBE has been known for decades, there are currently still no effective therapeutic drugs and preventive vaccines for TBE.” I must disagree; vaccines against TBEV are commonly available in Europe and Russia. The first vaccine was developed in 1937 in the former Soviet Union and was derived from mouse brain but resulted in frequent adverse effects [1]. Later, in the early 1970s, the first tissue culture-derived TBEV vaccine was developed [2]. Currently, there are several widely used vaccines (Encepur N, FSME-Immun CC and Ticovac, Encevir-Neo, and  Klesh-E-Vak) [1,2]. Hence, vaccination remains an effective way to prevent TBE.

Response 1: Thank you for your professional suggestions. We agree with this comment. The existing research literature indicates that there are successful vaccines for TBE prevention, but there is still no specific therapy for TBE infection, treatment approaches primarily focus on symptomatic relief and support[1-2]. Therefore, We have corrected the" there are currently still no effective therapeutic drugs and preventive vaccines for TBE " to " currently, there are still no specifically therapeutic drugs for TBE. " in the manuscript and marked it in red font. Relative information has been modified in lines 47-48.

References

[1]Taba P, Schmutzhard E, Forsberg P, Lutsar I, Ljøstad U, Mygland Å, Levchenko I, Strle F, Steiner I. EAN consensus review on prevention, diagnosis and management of tick-borne encephalitis. Eur J Neurol. 2017 Oct;24(10):1214-e61. doi: 10.1111/ene.13356. Epub 2017 Aug 1. PMID: 28762591. IF: 5.1 Q1

[2]Pustijanac E, Buršić M, Talapko J, Škrlec I, Meštrović T, Lišnjić D. Tick-Borne Encephalitis Virus: A Comprehensive Review of Transmission, Pathogenesis, Epidemiology, Clinical Manifestations, Diagnosis, and Prevention. Microorganisms. 2023 Jun 22;11(7):1634. doi: 10.3390/microorganisms11071634. PMID: 37512806; PMCID: PMC10383662. IF: 4.5 Q2

Comments 2: Line 142: “Real-time quantitative PCR (RT-qPCR)”. Please pay attention to using the proper names of molecular techniques. The “RT-PCR” should only be used to describe reverse transcription PCR and not the real-time PCR [Bustin, 2009]. Herein, since RNA was your matrix, reverse transcription was used to generate cDNA, and further quantitative amplification was applied; the proper name should be quantitative reverse transcription polymerase chain reaction (RT-qPCR or qRT-PCR) or real-time reverse transcription PCR (abbrev also RT-qPCR or qRT-PCR). Interestingly, herein, you have used both two-step and one-step approaches to RT-qPCR.

Response 2: Thank you for your professional suggestions. We agree with this comment. In the text, we have used two-step or one-step approach to RT-qPCR in different experiments. A one-step approach utilizing the Taqman probe was employed for detecting viral mRNA, while a two-step approach employing the SYBR green dye was utilized for identifying differential expressed genes following transcriptome sequencing. And we have corrected the" Real-time PCR " to " RT-qPCR. " in the manuscript and marked it in red font. Relative information has been modified in lines 142, 145.

Comments 3: Unfortunately, minocycline's antiviral potential was significantly dependent on the infectious dose of TBEV, as shown in Figure 3A (10-fold vs 100-fold TCID50). This is an important limitation and should be underlined and commented on more in the discussion.

Response 3: Thank you for your professional suggestions. We agree with this comment. Therefore, we added the interrelated comment in the discussion. Relative information has been modified in lines 379-380.

Reviewer 3 Report

Comments and Suggestions for Authors

The manuscript submitted by Cao et al., describes a drug screen assay evaluating the efficacy of minocycline at reducing TBEV infection in Vero cells.  Once minocycline was shown to have some level of inhibitory capacity, the authors evaluated the transcriptomic profile of TBEV infected cells in the presence and absence of minocycline.  Genes of particular interest were further evaluated by targeted PCR. The rationale for this work is that minocycline as been shown to reduce infection in cell culture by other flaviviruses, specifically West Nile, Japanese encephalitis and dengue viruses.  The intent of this effort is to gain an understanding of the mechanisms behind minocycline inhibition of TBEV infection.  While the scope and general approach of the project is reasonable, the presentation is poor with a litany of typographical and grammatical errors that make the text difficult to follow.  In addition, there are two fundamental flaws in this effort. The first is that the pathways that were identified in this study appear to be those already shown to be activated by minocycline meaning that there is really nothing novel here.  The second is that the latter studies evaluating targeted gene expression are lacking a critical minocycline only control that might demonstrate that the gene expression in TBEV infected cells is no different from drug only cells.  This same control should have been included in the transcriptomic analysis but I can appreciate the added cost for this effort. The authors need to repeat the studies shown in figures 7, 8 and 9 with a minocycline only control before proper interpretation of the data can be made.

Specific comments:

I have a lot of comments regarding manuscript presentation that I am not going to add here.  The authors need to complete a VERY thorough proofing of this manuscript.

Line 46:  TBEV vaccines have been available for decades in both Europe and Russia.  There are two inactivated virus vaccines in Europe alone.

Line 70:  Interesting information about ‘Verda Reno’ cells, but nobody calls them that and the Esperanto language is not common.  Stick to Vero cells.

Line 105:  Why do you do freeze thaw cycles for TBEV?  Mature TBEV is released into the cell culture supernatant, typically at high titers, although maybe not from Vero cells.  We use BHK cells.  The freeze thaw will release a lot of cell debris and immature particles that could confound your studies.

Line 142:  Why not use a NP gene standard so that PCR data can be specifically quantified?  Comparison against housekeeping genes is not quantitative and can be misleading.

Lines 158-159:  Should be DEGs

Somewhere in your methods the drug evaluation assays need to be clearly described.  For example, how long were the cells pretreated before adding the virus?  How long after infection was drug added.  Having this information in the figures alone is insufficient and also unclear.

Lines 225, 226 and 239:  Qualitative assessment of fluorescence intensity can’t be ‘significant’.

Section 3.3:  No information of how the drug treatments were performed.

Lines 273-275:  Don’t use “etc”.  It is meaningless here.

Sections 3.4-3.6:  Without proper controls, these data are uninterpretable.

Lines 395-400:  You are not measuring calcium homeostasis so any suggestion of impacting or restoring calcium homeostasis is inappropriate.  The impact of virus infection on calcium driven regulation of intracellular signaling is a long overlooked aspect of virus/cell biology.  If you were inclined to follow up on this, I suspect the results could be interesting.

Comments on the Quality of English Language

Significant editing is required both for grammar and general language use.

Author Response

Dear Editors and Reviewers:

Thank you for your letter and for the reviewers, comments concerning our manuscript entitled “Minocycline Inhibits Tick-Borne Encephalitis Virus and Protects Infected Cells via Multiple Pathways”(ID: viruses-3015788). Those comments are all valuable and very helpful for revising and improving our paper, as well as the important guiding significance to our researches. We have studied comments carefully and have made correction which we hope meet with approval. Revised portion are marked in red in the manuscript. The main corrections in the paper and the responds to the reviewer's comments areas flowing:

Specific comments:

Comments 1: Line 46:  TBEV vaccines have been available for decades in both Europe and Russia.  There are two inactivated virus vaccines in Europe alone.

Response 1: Thank you for your professional suggestions. We agree with this comment. The existing research literature indicates that there are successful vaccines for TBE prevention, but there is still no specific therapy for TBE infection, treatment approaches primarily focus on symptomatic relief and support[1-2]. Therefore, We have corrected the" there are currently still no effective therapeutic drugs and preventive vaccines for TBE " to " currently, there are still no specifically therapeutic drugs for TBE. " in the manuscript and marked it in red font. Relative information has been modified in lines 47-48.

References

[1]Taba P, Schmutzhard E, Forsberg P, Lutsar I, Ljøstad U, Mygland Å, Levchenko I, Strle F, Steiner I. EAN consensus review on prevention, diagnosis and management of tick-borne encephalitis. Eur J Neurol. 2017 Oct;24(10):1214-e61. doi: 10.1111/ene.13356. Epub 2017 Aug 1. PMID: 28762591. IF: 5.1 Q1

[2]Pustijanac E, Buršić M, Talapko J, Škrlec I, Meštrović T, Lišnjić D. Tick-Borne Encephalitis Virus: A Comprehensive Review of Transmission, Pathogenesis, Epidemiology, Clinical Manifestations, Diagnosis, and Prevention. Microorganisms. 2023 Jun 22;11(7):1634. doi: 10.3390/microorganisms11071634. PMID: 37512806; PMCID: PMC10383662. IF: 4.5 Q2

Comments 2: Line 70:  Interesting information about ‘Verda Reno’ cells, but nobody calls them that and the Esperanto language is not common.  Stick to Vero cells.

 Response 2: Thank you for your professional suggestions. We agree with this comment. Therefore, we have corrected " Verda Reno (Vero) cells " to "Vero" and marked it in red font. Relative information has been modified in line 70.

Comments 3:Line 105:  Why do you do freeze thaw cycles for TBEV?  Mature TBEV is released into the cell culture supernatant, typically at high titers, although maybe not from Vero cells.  We use BHK cells.  The freeze thaw will release a lot of cell debris and immature particles that could confound your studies.

 Response 3: Thank you for your professional suggestions. We agree with this comment. Therefore, to eliminate the effects of non-infectious intracellular virions, we redesigned the experiment, the collected supernatant was diluted directly in gradient and then infected cells. Lines 105-107 and the results in Figure 4C have been modified.

Comments 4: Line 142:  Why not use a NP gene standard so that PCR data can be specifically quantified?  Comparison against housekeeping genes is not quantitative and can be misleading.

Response 4: Thank you for your professional suggestions. We understand your concern, but our purpose is not to quantify the NP gene, but to observe the effect of Minocycline on the NP gene, we set up a control group and use a semi-quantitative method to obtain this result, and at the same time, we can use the same samples for the transcriptome validation analysis, to avoid experimental errors caused by different batches of samples.

Comments 5: Lines 158-159:  Should be DEGs

Response 5: Thank you for your professional suggestions. We have corrected " DGEs " to "DEGs" and marked it in red font. Relative information has been modified in line 159-160.

Comments 6: Somewhere in your methods the drug evaluation assays need to be clearly described.  For example, how long were the cells pretreated before adding the virus?  How long after infection was drug added.  Having this information in the figures alone is insufficient and also unclear.

Response 6: Thank you for your professional suggestions. We added the description about methods the drug evaluation assays. Relative information has been modified in lines 154-155, 167-168. Other experiments of drug evaluation methods were described in the results section, relative information has been modified in lines 217-218, 221-222, 234-236.

Comments 7: Lines 225, 226 and 239:  Qualitative assessment of fluorescence intensity can’t be ‘significant’.

Response 7: Thank you for your professional suggestions. We have corrected the description of fluorescence intensity. Relative information has been modified in lines 228-229,242.

Comments 8: Section 3.3:  No information of how the drug treatments were performed.

Response 8: Thank you for your professional suggestions. We added drug treatments information and marked it in red font. Relative information has been modified in lines 253-254.

Comments 9: Lines 273-275:  Don’t use “etc”.  It is meaningless here.

Response 9: Thank you for your professional suggestions. We have deleted " etc ". Relative information has been modified in lines 277-279.

Comments 10: Sections 3.4-3.6:  Without proper controls, these data are uninterpretable.

Response 10: Thank you for your professional suggestions. We agree with this comment. Therefore, we repeated the Sections 3.4-3.6 experiment with an additional 20 µM MIN treatment group. The experimental results indicated that the mRNA level of CALML4, RYR2, SNAP25, FGF2, PDGFRA, PLCB2 and SYT11 exhibited no significant difference compared to the control group, however, the mRNA level of IL-6 showed a significant decrease and the IL-6 in the supernatant also demonstrated a decreasing trend without statistical significance. The relevant experimental results were shown in fig7B, fig7C, fig8A, fig8B, fig9A, fig9B, fig9C, respectively. Relative information has been modified in lines 317-319, 332-333, 337-338, 364-366

Comments 11: Lines 395-400:  You are not measuring calcium homeostasis so any suggestion of impacting or restoring calcium homeostasis is inappropriate.  The impact of virus infection on calcium driven regulation of intracellular signaling is a long overlooked aspect of virus/cell biology.  If you were inclined to follow up on this, I suspect the results could be interesting.

Response 11: Thank you for your professional suggestions. We agree with this comment. Therefore, we have corrected " calcium homeostasis " to " expression of calcium signaling pathway associated proteins " and marked it in red font. Relative information has been modified in lines 409-410.

I agree with your comments about the relationship between calcium ions and viral replication, and I also think it is an interesting topic to study it.

Round 2

Reviewer 1 Report

Comments and Suggestions for Authors

The article can be published.

Author Response

Dear Reviewer,

Thank you for your letter and for agreeing to publish our article entitled “Minocycline Inhibits Tick-Borne Encephalitis Virus and Protects Infected Cells via Multiple Pathways”(ID: viruses-3015788) in the journal viruses.

yours sincerely

Chunyuan Li

Reviewer 2 Report

Comments and Suggestions for Authors

Dear Authors,

Thank you for responding to my previous comments. However, I must request one additional correction. Previously, I had commented on the vaccines against TBEV. You have corrected this part in the updated version of the manuscript, yet I was hoping that you also add information about vaccination. The whole manuscript focuses on TBEV and TBE as major public health threats, but there is no information that it is a preventable disease. This is a major oversight! Please include detailed information on the specific prevention of TBE, according to my previous comments.

Comments on the Quality of English Language

The text would benefit from some minor language and style revisions to improve its clarity and coherence.

Author Response

Dear Editors and Reviewers:

Thank you for your letter and for the reviewers, comments concerning our manuscript entitled “Minocycline Inhibits Tick-Borne Encephalitis Virus and Protects Infected Cells via Multiple Pathways”(ID: viruses-3015788). Those comments are all valuable and very helpful for revising and improving our paper, as well as the important guiding significance to our researches. We have studied comments carefully and have made correction which we hope meet with approval. Revised portion are marked in purple in the manuscript. The main corrections in the paper and the responds to the reviewer's comments areas flowing:

Comments 1: Thank you for responding to my previous comments. However, I must request one additional correction. Previously, I had commented on the vaccines against TBEV. You have corrected this part in the updated version of the manuscript, yet I was hoping that you also add information about vaccination. The whole manuscript focuses on TBEV and TBE as major public health threats, but there is no information that it is a preventable disease. This is a major oversight! Please include detailed information on the specific prevention of TBE, according to my previous comments.

Response 1: Thank you for your professional suggestions. We added the comments about the information about vaccination and marked it in purple font. Relative information has been modified in lines 47-58.

Reviewer 3 Report

Comments and Suggestions for Authors

The revised submission by Cao et al., is improved over the initial submission.  Most of my concerns have been addressed adequately and the performance of additional experiments to include proper controls improves the quality of the data and provides better support for the conclusions the authors have drawn.

I do still have some minor comments:

Line 73: Which strain of TBEV was used in this work? Assuming Far-Eastern subtype.

Figure legends: Please add language for the statistical notations.  Copy/paste from lines 178-179, as appropriate would work

Line 253: Should be 'DEGs" rather than DGEs.

Line 306: change to "...MIN could potentially impact calcium homeostasis..."

Line 310:  Please indicate what is the 'control group'.  Assuming uninfected/untreated cells.

Lines 318, 333 and 338: 'Difference' should be 'different'

Comments on the Quality of English Language

Minor grammatical errors need to be addressed

Author Response

Dear Editors and Reviewers:

Thank you for your letter and for the reviewers, comments concerning our manuscript entitled “Minocycline Inhibits Tick-Borne Encephalitis Virus and Protects Infected Cells via Multiple Pathways”(ID: viruses-3015788). Those comments are all valuable and very helpful for revising and improving our paper, as well as the important guiding significance to our researches. We have studied comments carefully and have made correction which we hope meet with approval. Revised portion are marked in purple in the manuscript. The main corrections in the paper and the responds to the reviewer's comments areas flowing:

Comments 1: Line 73: Which strain of TBEV was used in this work? Assuming Far-Eastern subtype.

Response 1: Thank you for your professional suggestions. We added the TBEV subtype, "European subtype", and marked it in purple font. Relative information has been modified in line 85.

Comments 2: Figure legends: Please add language for the statistical notations.  Copy/paste from lines 178-179, as appropriate would work

Response 2: Thank you for your professional suggestions. We added the language for the statistical notations, and marked it in purple font. Relative information has been modified in lines 216-218, 231-233, 247-248, 264-265, 341-344, 369-372 and 397-400.

Comments 3: Line 253: Should be 'DEGs" rather than DGEs.

Response 3: Thank you for your professional suggestions. We have corrected the "DGEs" to "DEGs" in the manuscript and marked it in purple font. Relative information has been modified in line 269.

Comments 4: Line 306: change to "...MIN could potentially impact calcium homeostasis..."

Response 4: Thank you for your professional suggestions. We have corrected the "MIN restores calcium homeostasis" to " MIN could potentially impact calcium homeostasis " in the manuscript and marked it in purple font. Relative information has been modified in line 322.

Comments 5: Line 310: Please indicate what is the 'control group'.  Assuming uninfected/untreated cells.

Response 5: Thank you for your professional suggestions. We have corrected the " Compared with the control group " to " Compared with the Vero cells group " in the manuscript and marked it in purple font. Relative information has been modified in lines 327, 330, 334, 353, 358 and 389. In 3.4-3.6 parts of  this manuscript, the control group was Vero cell without virus infection or drug treatment.

Comments 6: Lines 318, 333 and 338: 'Difference' should be 'different'

Response 6: Thank you for your professional suggestions. We have corrected the " Difference " to " different" in the manuscript and marked it in purple font. Relative information has been modified in line 335, 354 and 359.
